# Ultrahigh resistance of hexagonal boron nitride to mineral scale formation

Kuichang Zuo [1,2,3,10], Xiang Zhang [3,4,10], Xiaochuan Huang [2,3], Eliezer F. Oliveira[4,5], Hua Guo[4], Tianshu Zhai[4], Weipeng Wang[6] ✉, Pedro J. J. Alvarez [2,3], Menachem Elimelech [3,7], Pulickel M. Ajayan [3,4] ✉, Jun Lou [3,4,8] ✉ & Qilin Li [2,3,4,9] ✉

Formation of mineral scale on a material surface has profound impact on a wide range of natural processes as well as industrial applications. However, how specific material surface characteristics affect the mineral-surface interactions and subsequent mineral scale formation is not well understood. Here we report the superior resistance of hexagonal boron nitride (hBN) to mineral scale formation compared to not only common metal and polymer surfaces but also the highly scaling-resistant graphene, making hBN possibly the most scaling resistant material reported to date. Experimental and simulation results reveal that this ultrahigh scaling-resistance is attributed to the combination of hBN's atomically-smooth surface, in-plane atomic energy corrugation due to the polar boron-nitrogen bond, and the close match between its interatomic spacing and the size of water molecules. The latter two properties lead to strong polar interactions with water and hence the formation of a dense hydration layer, which strongly hinders the approach of mineral ions and crystals, decreasing both surface heterogeneous nucleation and crystal attachment.

Interfacial interactions play a fundamental role in many aqueous processes including adsorption, catalytic reaction, corrosion, filtration, and scale formation. In particular, scale formation, i.e., the development of mineral deposits on a material surface due to precipitation from the bulk solution and/or crystal formation initiated by surface nucleation, has great impacts on interfacial transfer of mass, heat, electrons, and light. It causes profound performance decline in numerous industrial processes, such as impaired heat transfer in heat exchangers and boilers, increased pressure drop in pipes, flow blockage in filtration membranes, corrosion damage of steam turbines, decreased conductivity and activity of electrodes, premature failure of heating and electrochemical components etc.[1–4], all leading to higher operation cost and safety risk. It is reported that economical loss due to mineral scaling in boilers, turbines, and heat exchangers accounts for 0.17–0.25% of the gross domestic product (GDP) in industrialized nations[5]. Understanding mineral scaling behavior is important for the development of next generation materials and technologies that address these critical challenges.

[1]The Key Laboratory of Water and Sediment Sciences, Ministry of Education; College of Environment Sciences and Engineering, Peking University, Beijing 100871, China. [2]Department of Civil and Environmental Engineering, Rice University, MS 519, 6100 Main Street, Houston, TX 77005, USA. [3]NSF Nanosystems Engineering Research Center Nanotechnology-Enabled Water Treatment, Rice University, MS 6398, 6100 Main Street, Houston, TX 77005, USA. [4]Department of Materials Science and NanoEngineering, Rice University, 6100 Main Street, Houston, TX 77005, USA. [5]São Paulo State Department of Education, São Paulo, Brazil. [6]Key Laboratory of Advanced Materials (MOE), School of Materials Science and Engineering, Tsinghua University, Beijing 100084, PR China. [7]Department of Chemical and Environmental Engineering, Yale University, New Haven, CT 06520-8286, USA. [8]Department of Chemistry, Rice University, 6100 Main Street, Houston, TX 77005, USA. [9]Department of Chemical and Biomolecular Engineering, Rice University, 6100 Main Street, Houston, TX 77005, USA. [10]These authors contributed equally: Kuichang Zuo, Xiang Zhang. ✉e-mail: wpwang@mail.tsinghua.edu.cn; ajayan@rice.edu; jlou@rice.edu; qilin.li@rice.edu

Scale formation can occur through the deposition of mineral crystals formed in the bulk solution, as well as through surface-induced heterogeneous nucleation with crystals growing from nucleation sites on a surface[6]. Both processes are strongly influenced by material surface properties. Similar to deposition of other particles, material properties affect the attachment of mineral crystals via hydrophobic and electrostatic interactions. Surface-induced heterogeneous nucleation is a more thermodynamically favorable process, but it is poorly understood because it happens on very small time and length scales[7]. Few previous studies have investigated the different surface properties that influence surface-induced heterogeneous nucleation: roughness, charge, and hydrophobicity[6]. Surface roughness is directly related to the number of nucleation sites; it is generally recognized that mineral crystallization increases with surface roughness. However, findings on the impact of charge and surface hydrophobicity have been inconsistent. For example, some studies found that surface charge influenced the heterogeneous nucleation via electrostatic interactions or complexation reactions with the mineral ions[2,8], while others reported similar nucleation rates on surfaces of different charges[9]. Contradictory results have also been reported on the role of surface hydrophobicity in mineral scaling. Hydrophilic coatings such as graphene oxide (GO), graft polymers, and polyethylene glycol, were shown to delay the onset of $CaCO_3$ scaling in some studies[10–12], while other studies showed that hydrophilic surfaces promoted $CaCO_3$ nucleation[8], and GO had no anti-scaling effect[13]. One possible reason for such apparent contradiction is that modification of a surface property (e.g., hydrophobicity or charge) often leads to inevitable changes in other surface properties, which makes it difficult to discern the role of an individual surface property for development of anti-scaling materials.

Two dimensional (2D) materials possess atomically smooth surfaces, and have attracted tremendous interest for their potential applications in processes where interfacial interactions play a critical role. For example, graphene was shown to effectively inhibit nucleation of metals[14] and metal oxides[15] in nonaqueous phases. Theoretical and experimental research on water-surface interactions[16–21] and related phenomena, such as ultrafast water transport[22–26], de-icing[27], and anti-fouling[28], also suggest that the atomically smooth morphology plays an important role in preventing adhesion. However, no studies have investigated the scaling behavior, especially heterogeneous nucleation, on the surface of 2D nanomaterials such as graphene and hexagonal boron nitride (hBN) in aqueous solutions. Furthermore, 2D materials vary widely in surface chemistry despite their common feature of atomic-level smoothness: Graphene consists of a single layer of carbon atoms arranged in a honeycomb lattice nanostructure, featuring a small lattice constant, low in-plane polarity, and high hydrophobicity; hBN, another important 2D material with its lattice structure and lattice constant similar to those of graphene[29,30], has high in-plane polarity due to the boron-nitrogen bond and higher hydrophilicity than graphene[18]. It is unknown how such differences in surface chemistry influence mineral scale formation on these atomically smooth surfaces.

Here we investigate mineral scale formation on graphene and hBN surface, and compare it to that on metal (i.e., titanium (Ti)) and polymer (i.e., polyvinylidene fluoride (PVDF)) surfaces commonly used for high scaling potential applications (Supplementary Note 1, Supplementary Figs. 1 and 2). Surface induced heterogeneous nucleation on theses surfaces are investigated using both real time, in situ measurements as well as ex situ characterization methods. We also directly quantify the binding force of mineral crystals grown from surface-induced heterogenous nucleation. Experimental measurements combined with density functional theory (DFT) illustrate the effect of surface chemistry on the formation of the hydration layer and its key role in mineral ion-surface interactions. Very importantly, we discover hBN's outstanding anti-scaling properties and demonstrate its potential application as an anti-scaling coating in practical engineering systems.

## Results

### Anti-scaling behavior of hBN

To evaluate the scaling behaviors of graphene and hBN, graphene and hBN nanocoatings grown on flat Cu substrate were dipped in supersaturated $CaSO_4$ solution (50 mmol $L^{-1}$, saturation index (SI) of 3.28) and characterized using a video camera for 220 min (Fig. 1A). As the induction time of homogeneous nucleation for $CaSO_4$ at this concentration (<40 min[6]) is much shorter than the operation time, both bulk precipitation due to homogeneous nucleation and surface induced heterogeneous nucleation contribute to scale formation. As

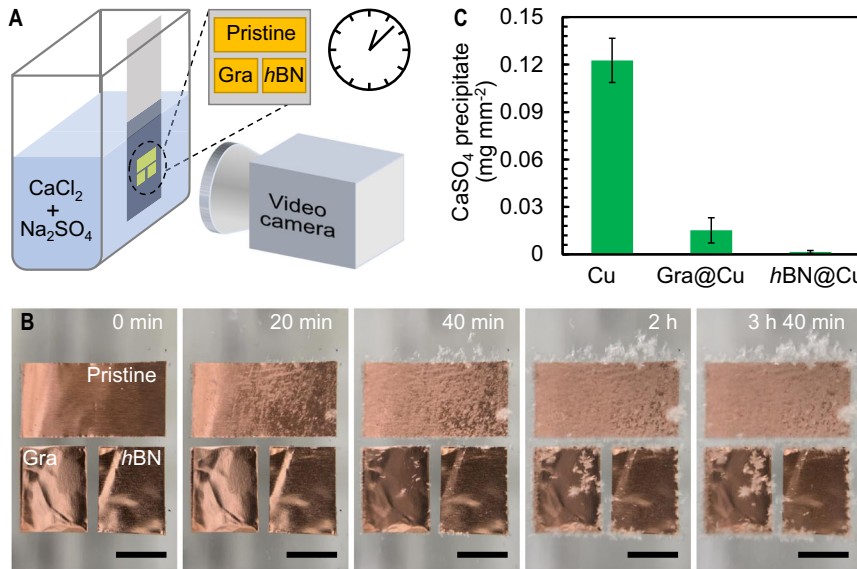

**Fig. 1 | Scaling behavior of graphene and hBN caused by both surface-induced heterogeneous nucleation and attachment of mineral crystals formed in the bulk solution through homogeneous nucleation. A** Experimental setup and **B** video snapshots of the pristine Cu (pristine), graphene (Gra) coated Cu (Gra@Cu), and hBN coated Cu (hBN@Cu) exposed to a supersaturated $CaSO_4$ solution. The scale bars are 3 mm in length; **C** Amount of $CaSO_4$ precipitate formed on the three samples after 220 min of exposure. Error bars represent the standard deviation of precipitation amount.

shown in Fig. 1B and Supplementary Movie 1, $CaSO_4$ crystals were found on the pristine Cu surface after 20 min, and severe scale formation ($0.123 \pm 0.014$ mg mm$^{-2}$) was observed after 220 min of testing. On the graphene surface, $CaSO_4$ crystals occurred after 40 min, and increased to $0.015 \pm 0.008$ mg mm$^{-2}$ at the end of the experiment (Fig. 1C). However, the $h$BN-coated Cu surface exhibited almost no scale formation ($0.001 \pm 0.001$ mg mm$^{-2}$) during the 220 min experiment, except at its edges where the $h$BN nanocoating was damaged due to sample cutting (Fig. 1B).

After illustrating the macroscopic anti-scaling behavior of graphene and $h$BN using video camera, we investigated the scale formation caused by surface-induced heterogeneous nucleation on graphene, $h$BN, Ti, and PVDF using microscopic ex situ and in situ measurements with supersaturated $CaCO_3$. $CaCO_3$ and $CaSO_4$ are both common scalants in water, but $CaCO_3$ has a solubility (0.12 mmol L$^{-1}$) two orders of magnitude lower than $CaSO_4$ (19.10 mmol L$^{-1}$). It allows using much lower concentration to prepare supersaturated $CaCO_3$ solution with saturation index similar to supersaturated $CaSO_4$, which would generate smaller amount of scalants on the surfaces and is conducive to performing microscopic ex situ or in situ characterizations. As shown in Fig. 2A, supersaturated $CaCO_3$ (SI of 1.18) was introduced to the test surfaces immediately after preparation before homogeneous nucleation occurred (Supplementary Note 2, Supplementary Fig. 3). Vaterite was the main $CaCO_3$ crystal formed on the test surfaces (Supplementary Fig. 4). Its growth was the fastest on Ti (Fig. 2B), followed by PVDF (Fig. 2C), graphene (Fig. 2D), and $h$BN (Fig. 2E). The same trend was observed at different reaction time (Fig. 2F). In over 15 h, almost no vaterite was observed on $h$BN

($0.4 \pm 0.2$ mm$^{-2}$), while $55.6 \pm 3.3$, $13.2 \pm 4.7$, and $3.6 \pm 0.8$ vaterite crystals mm$^{-2}$ were found on Ti, PVDF, and graphene surfaces, respectively (Fig. 2G). The very few vaterite crystals formed on $h$BN were much smaller in size than those formed on the other surfaces. These results suggest prolonged vaterite heterogeneous nucleation induction time and slower vaterite formation kinetics on the $h$BN surface.

Real time, in situ measurement of $CaCO_3$ nucleation and growth rates using the quartz crystal microbalance with dissipation (QCMD) technique (Fig. 2A) further confirms that $h$BN greatly hinders surface-induced heterogeneous nucleation of $CaCO_3$ (Fig. 2H, I). Upon introduction of an influent solution supersaturated with $CaCO_3$, adsorption of ions (e.g., $Ca^{2+}$, $CO_3^{2-}$) led to an immediate albeit small change in frequency and dissipation, which reached equilibrium in ~1 h. The larger frequency and dissipation changes observed for the Ti and PVDF surfaces suggested more adsorption of $Ca^{2+}$ and $CO_3^{2-}$ ions than that on the graphene and $h$BN surfaces. The similar frequency and dissipation changes on graphene and $h$BN surface suggested that they had similar ion adsorption, a result of their similar surface charge[31,32]. After the initial ion adsorption, greater frequency and dissipation changes occurred, and exhibited two distinct stages. The first stage (stage i) featured relatively slow changes in frequency (dF/dt) and dissipation (dD/dt), signaling the formation of vaterite nuclei, i.e., the induction process. This was followed by a much faster frequency shift (stage ii), which signaled crystal growth from the nuclei previously formed on the surface (Fig. 2I). $CaCO_3$ mass accumulation rate calculated using the Sauerbrey equation (Supplementary Fig. 5) showed a very large increase in stage ii compared to that in stage i on Ti, PVDF

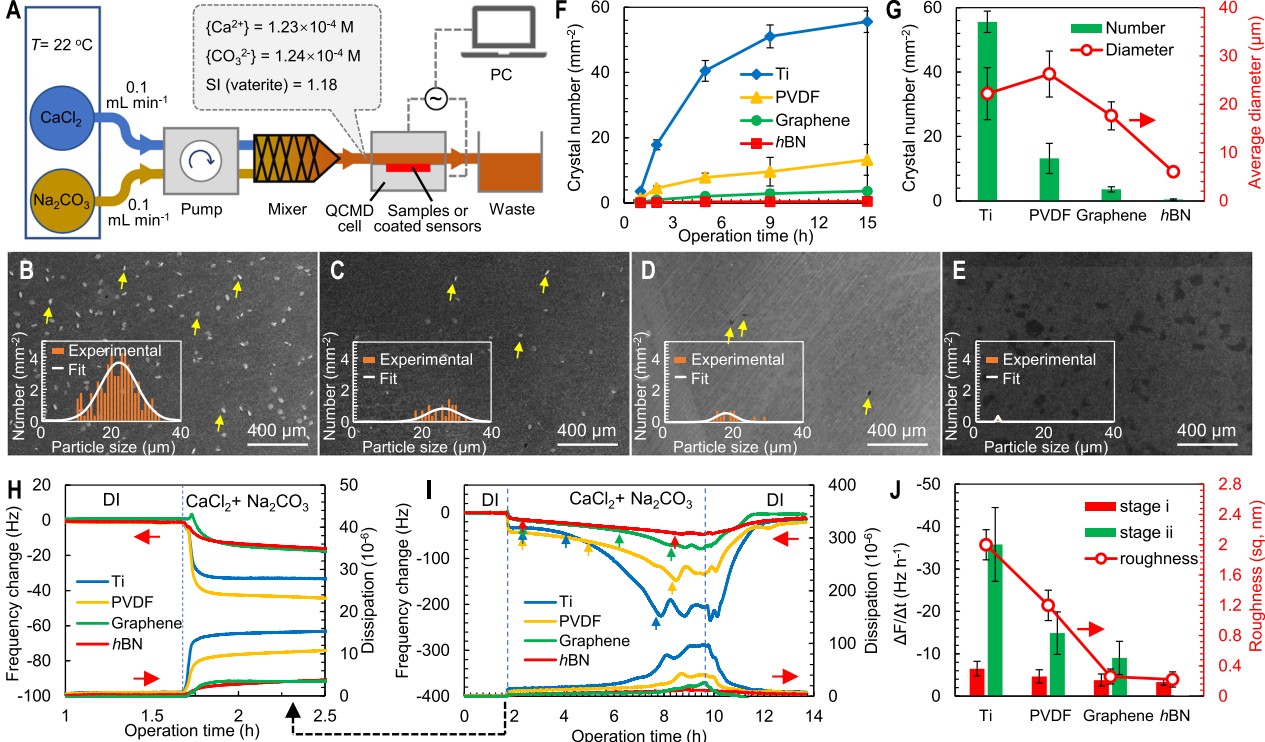

**Fig. 2 | Characterization of surface-induced heterogeneous nucleation of $CaCO_3$ on Ti, PVDF, graphene, and $h$BN. A** A schematic of QCMD experimental setup for characterizing $CaCO_3$ heterogeneous nucleation on four surfaces. SI: saturation index, PC: personal computer, QCMD: quartz crystal microbalance with dissipation; SEM images of $CaCO_3$ crystals formed on **B** Ti, **C** PVDF, **D** graphene, and **E** $h$BN coatings on Cu substrate after 9 h of operation, with insets showing the size distribution of vaterite crystals formed. The particle size distribution shown in the B–E inserts are obtained using the ImageJ software and fitted with a Gaussian distribution function; Average number and size of vaterite crystals grown on the four surfaces (**F**) as a function of operation time and **G** after 15 h of operation; **H**–**J** Real time in-situ monitoring of $CaCO_3$ scale formation using QCMD. **H** Ion adsorption. **I** Induction and growth of $CaCO_3$ crystals. Three arrows marked on the Ti, PVDF, and graphene curves indicate the start and finish of two stages: the induction stage i and the growth stage ii. Data obtained using the $h$BN sample only exhibit one stage (stage i) during the operation. **J** Average frequency change rate ($\Delta F/\Delta t$) during the two-stage scale-formation process and surface roughness of the four nanocoatings. Error bars represent the standard deviation.

and graphene surfaces. As shown in Fig. 2I, J, the Ti surface showed the shortest induction time and the fastest crystal growth kinetics, followed by PVDF. The heterogeneous nucleation induction time on graphene was significantly longer compared to Ti and PVDF (Fig. 2I), and a much slower crystal growth kinetics was observed (Fig. 2J and Supplementary Fig. 5). On the $h$BN surface, the frequency and dissipation shifts throughout the experiment were very small. It did not exhibit a distinct stage ii with fast frequency and dissipation changes. This suggests that the induction time on the $h$BN surface is longer than the approximately 8 h contact time with the test solution in the experiment; no detectable crystal growth occurred during this period (Fig. 2I, J and Supplementary Fig. 5). This may be attributed to the low ion adsorption on $h$BN (Fig. 2I), which prolongs the induction of heterogeneous nucleation and hinders vaterite crystal growth. After 10 h of operation, the system was flushed with ultrapure water. Frequency and dissipation signals approached the original DI water baseline at different rate for all samples (Fig. 2I), confirming that the frequency and dissipation changes observed earlier resulted from the nucleation and growth of vaterite, which was removed or dissolved when flushed with ultrapure water.

## Atomic smoothness decreases nucleation and binding

Although the four material surfaces are all nominally flat and smooth at the macroscopic length scale, their nanoscale morphologies differ significantly (Fig. 3A). The root-mean-square roughness values of Ti ($2.0 \pm 0.2$ nm) and PVDF ($1.2 \pm 0.2$ nm) are almost an order of magnitude higher than those of graphene ($0.3 \pm 0.1$ nm) and $h$BN ($0.2 \pm 0.1$ nm), which are atomically smooth with no observable sags, crests, or other surface irregularities. During the nucleation and crystal growth stages, the nanoscale rough features on Ti and PVDF serve as heterogeneous nucleation sites, provide cratered surface for nuclei attachment, and increases friction that may hinder local flow velocity[33], all conducive to vaterite nucleation and crystal growth, resulting in a shorter induction stage and faster vaterite growth (Fig. 2I, J).

During the crystal growth stage, the higher roughness of Ti and PVDF surfaces increases the contact area with the vaterite crystals, resulting in stronger binding forces. We quantify the lateral force required to detach vaterite crystals from the surface using a nanoindenter inside a SEM (Supplementary Note 3, Supplementary Fig. 6). The applied load increases linearly with the lateral travelling distance of the nanoindenter tip relative to the vaterite crystal, until the vaterite crystal is pushed off the surface (Fig. 3E–H, and Supplementary Movies 2 and 3), i.e., the "detaching point". The applied load at the detaching point is defined as the detaching force, which reflects the binding force between the vaterite crystal and the sample surface. The contact area (Fig. 3G) normalized detaching force followed the order of Ti ($44.9 \pm 12.7$ μN μm$^{-2}$) > PVDF ($27.6 \pm 3.4$ μN μm$^{-2}$) > $h$BN ($12.1 \pm 4.3$ μN μm$^{-2}$) ≈ graphene ($11.1 \pm 6.1$ μN μm$^{-2}$), correlating well with the surface roughness (Figs. 2J and 3I). The atomically smooth graphene and $h$BN exhibit notably lower binding forces. Note that these forces are much higher than typical colloidal adhesion forces[34], a notable distinction between scaling due to crystal formation from surface-induced heterogeneous nucleation and that due to deposition of mineral crystals formed in the bulk solution.

The lower binding force on the atomically-smooth graphene and $h$BN surfaces measured in our study is consistent with findings from previous studies that show atomically smooth carbon and boron-nitride materials exhibit super-lubricity at solid/solid rigid junctions[35–38] and peculiar interactions with water in 1D nanotubes[22,39–41] or 2D structures[21,23,24,42–44] at their solid/liquid interfaces. The reduced binding force allows easier detachment of the mineral crystals and decreases the formation of scales under hydraulic shear.

## Dense hydration layer mitigates nucleation

The similarity in their atomically smooth surface morphology is apparently responsible for the similarity in low vaterite detachment forces on graphene and $h$BN. The lower nucleation rate on $h$BN compared to graphene, however, suggests the difference in surface chemistry may play an important role. This is supported by DFT calculations. As a result of the delocalized π system in graphene, the charge distribution on graphene surface is much more uniform than on $h$BN, with a small dipole moment (0.036 D) and moderate carbon atom electronegativity (2.55) (Fig. 4A, B, Supplementary Fig. 7, Supplementary Note 4). In $h$BN, N atoms share their lone pair electrons with B, and the electrons of the π system are more localized on the N atoms. The electronegativity of N (3.04) is therefore notably higher than B (2.04), leading to corrugated charge distribution on $h$BN surface with a dipole moment of 0.047 D (Fig. 4A). The highest occupied molecular orbital (HOMO) is more localized on N atoms, and the

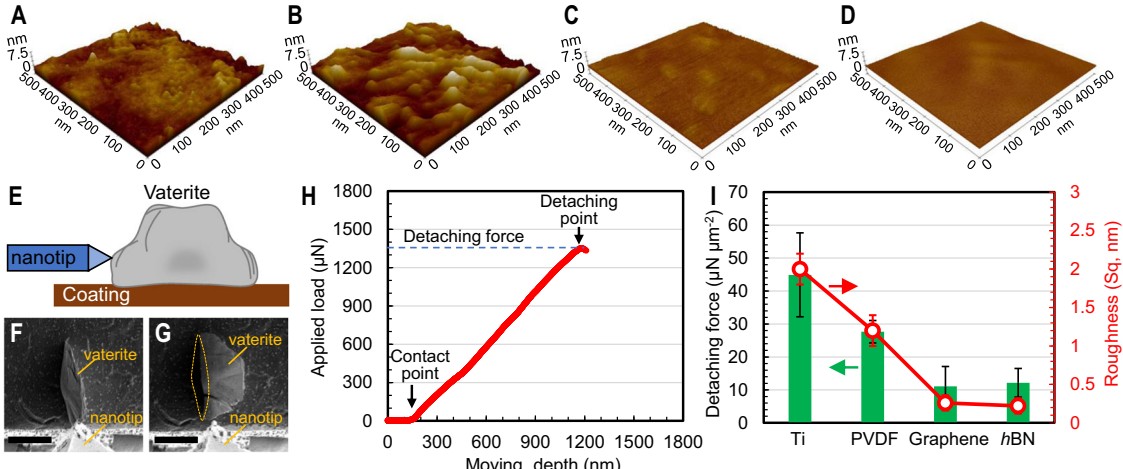

**Fig. 3 | Atomically-smooth morphology yields low scalant binding force.** AFM image of **A** Ti, **B** PVDF, **C** graphene, **D** $h$BN surfaces; **E**–**H** Measurement of vaterite detaching force from various coatings. **E** Sectional view schematic showing the measurement method by adding a lateral force on a vaterite crystal grown on a sample surface. Top view SEM image of a vaterite crystal (**F**) before and **G** after being pushed off from the sample surface by the nanoindenter tip. The scale bars in **F** and **G** represent 10 μm. The dashed circle in **G** shows the contact area between the vaterite crystal and the underlying sample surface after detachment; **H** Applied force load increases linearly with the moving depth of the nanotip upon contact to the vaterite crystal; **I** The average vaterite detaching force and surface roughness of the four nanocoatings. Error bars represent the standard deviation of corresponding results.

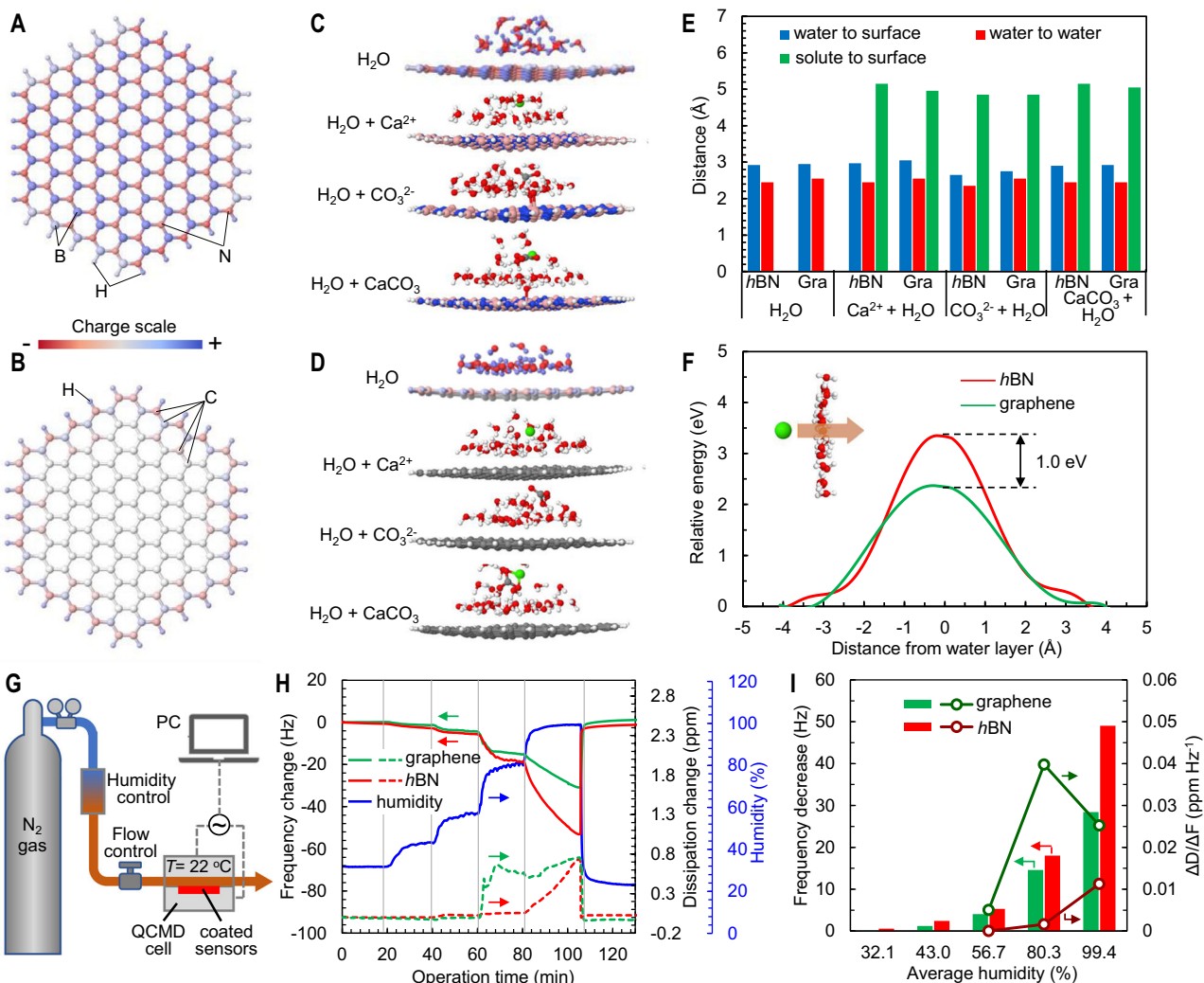

**Fig. 4 | Effect of surface chemistry on nucleation behavior of CaCO₃ on hBN and graphene surface.** DFT simulated charge distribution on **A** hBN and **B** graphene surfaces. Blue and red spheres in **A** represent boron and nitrogen atoms, respectively. Silver, light blue, and light red spheres in **B** represent carbon atoms (Supplementary Fig. 7). The edge of both hBN and graphene flakes are passivated with H atoms; DFT simulated interaction of H₂O, Ca²⁺, CO₃²⁻, and CaCO₃ with **C** hBN and **D** graphene surface (Supplementary Fig. 11); **E** DFT calculated distance between water and surface, between water molecules, and between solutes (Ca²⁺, CO₃²⁻, or CaCO₃) and hBN or graphene surfaces. Gra: graphene; **F** Minimum energy needed for a Ca²⁺ ion to penetrate the hydration layer on hBN and graphene surface calculated by DFT. Detailed calculation and simulation processes and results can be seen in Supplementary Note 4. **G**–**I** The formation of hydration layer on hBN and graphene surface characterized by QCMD. **G** QCMD experimental setup. PC: personal computer; **H** Frequency and dissipation change on hBN and graphene surfaces with changes in gas flow humidity; **I** The effect of gas flow humidity on average frequency decrease and the absolute ratio of dissipation change over frequency decrease (|ΔD/ΔF|).

lowest unoccupied molecular orbital (LUMO) is more concentrated on B atoms.

The non-uniform charge distribution of the hBN surface has an important influence on its interaction with polar water molecules. The alternating negative and positive charge regions on hBN attract H and O atoms in water, respectively, leading to stronger interaction with water molecules than on graphene surface, where the charge is more uniform. As a result, water molecules are more closely packed and located closer to the hBN surface: the average distance between water molecules is 2.45 Å, and the closest distance between water molecule and the hBN surface is around 2.92 Å (Fig. 4C–E and Supplementary Fig. 8), consistent with previously reported values from quantum Monte Carlo simulations[45]. In comparison, on graphene surface, the closest water molecules are located 2.95 Å away from the surface, and the distance between water molecules measures 2.55 Å on average. Similar results have been reported in previous studies. Utilizing ab initio molecular dynamics simulation, it was reported that the friction coefficient on hBN was about 3 times larger than that on graphene, and the adsorbed water molecule had faster slippage on graphene than on hBN[18].

The water molecule packing on the surface has direct impact on ion-surface interactions. Simulation of interactions between Ca²⁺, CO₃²⁻, or CaCO₃ with hBN or graphene in the presence and absence of water (Supplementary Figs. 9–11) show that hBN binds more strongly with Ca²⁺, CO₃²⁻, and CaCO₃ than graphene in the absence of water molecules (Supplementary Fig. 9), due to the locally charged B and N having higher attraction to the charged Ca²⁺, CO₃²⁻, and polar CaCO₃ species. However, when water molecules are introduced, the distance between the surface and Ca²⁺ or CaCO₃ is larger on hBN than on graphene (Fig. 4E and Supplementary Fig. 11). This is attributed to the denser hydration layer on hBN, which hinders the approach of Ca²⁺ and CaCO₃ to the surface. More energy (1.0 eV) is required for a Ca²⁺ ion to penetrate the hydration layer to reach the hBN surface than the graphene surface (Fig. 4F).

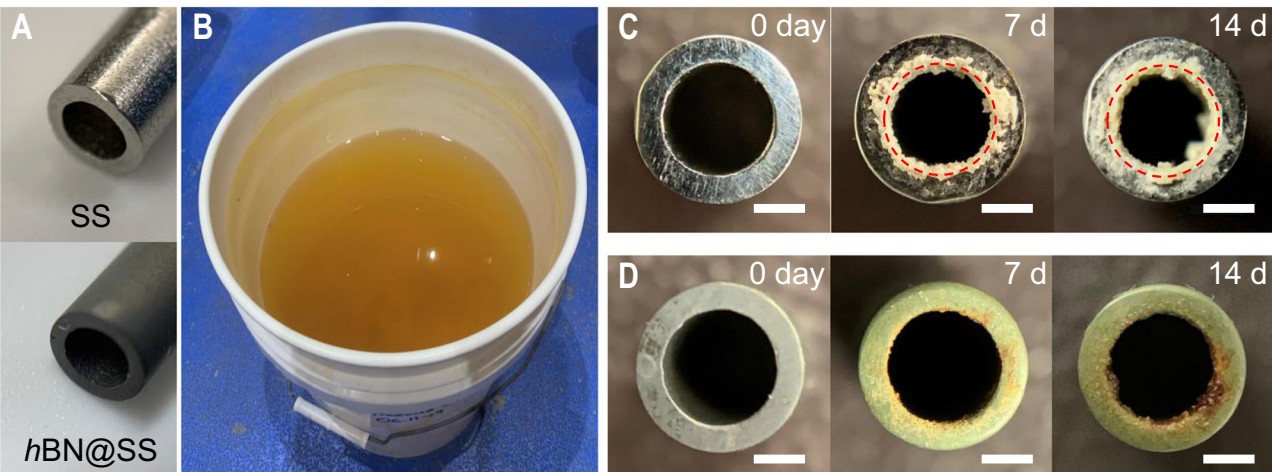

**Fig. 5 | Scaling mitigation of *h*BN nanocoating in real oil & gas produced water.**
**A** Photograph of stainless steel (SS) tube before (top) and after (bottom) the growth
of *h*BN nanocoating (*h*BN@SS); **B** A photograph of produced water taken from an
oil & gas production site in Texas, USA; Scaling behavior of the **C** pristine and **D** *h*BN

coated SS tube during 14 days of exposure to real produced water. The scale bars
are 1 mm in length. Red dash circles indicate the original inner diameter of the
pristine SS tube.

Experimental measurements of water vapor adsorption confirm
the simulation results. Adsorption of water vapor on graphene and
*h*BN surfaces was measured using the QCMD technique at varying
relative humidity of the influent $N_2/H_2O$ mixed gas (Fig. 4G). Both
graphene and *h*BN coated quartz sensors exhibit a decrease in the
resonate frequency with the increase in the relative humidity of the
influent gas (Fig. 4H), indicating increasing adsorption of water
molecules on graphene and *h*BN surfaces. The *h*BN coated sensor
experiences larger frequency decreases than the graphene coated
sensor at all humidity levels (Fig. 4H, I), suggesting more adsorption of
water on *h*BN than on graphene. Furthermore, the ratio of dissipation
increase over frequency change ($\Delta D/\Delta F$) is smaller on *h*BN than on
graphene (Fig. 4I). As water slippage on surfaces has little impact on $\Delta D$
and $\Delta F$[46,47], the measured $\Delta D/\Delta F$ directly reflects the viscoelastic
properties of the adsorbed water layer[48]. The smaller $\Delta D/\Delta F$ on *h*BN
indicates that the water layer adsorbed on the polar *h*BN surface is
more compact and viscous compared to that on the uniformly charged
graphene, consistent with the simulation results that a denser and
more compact water layer forms on *h*BN. Although several studies
have hinted the higher affinity of *h*BN for water[18,21,41], our study
experimentally confirm the denser and more rigid interfacial water
layer structure on *h*BN than graphene.

Condensation of water molecules due to van der Waals attraction
to form a hydration layer with higher density than bulk water has been
discussed for materials of the carbon and boron-nitride family[17,18,21],
nanotubes[39–41], and layered lamellar structures[23,42,49,50]. Our DFT simu-
lation and QCMD water vapor adsorption experiments demonstrate
the alignment of water molecules and formation of a condensed
hydration layer on *h*BN. Interestingly, the calculated distance between
water molecules on the 2D graphene and *h*BN surfaces (2.45–2.65 Å)
are smaller than those reported for conventional material surfaces,
such as NaCl(001) (-3.0 Å[51]), Cu(110) (2.8 ± 0.2 Å[52]), and Au(111)
(2.71–2.88 Å[53]), as well as the calculated intermolecular distance
between water molecules in bulk water (2.81 Å). The denser interfacial
water layer formed on the 2D ultra-smooth graphene and *h*BN surfaces
may be related to their small atom size and lattice constant that closely
matches the size of water molecule. Most conventional materials have
lattice constants larger than 3.0 Å (Supplementary Table 1), sig-
nificantly larger than the water molecule size (2.7 Å), causing loose
distribution of water on their surfaces. The lattice constants of gra-
phene (2.46 Å) and *h*BN (2.50 Å) match the water molecule size better,
with *h*BN's lattice constant being the closest to a water molecule size.

The close match between the distribution of *h*BN surface energy cor-
rugation (charge nonuniformity) and the di-pole of water molecules
leads to strong polar interactions between the *h*BN surface and polar
water molecules, and hence compact packing of water molecules on
*h*BN surface. Combining an atomically smooth surface, large in-plane
polarity, and a lattice constant matching best with the size of water
molecules (Supplementary Table 1), *h*BN is possibly one of the most
anti-scaling materials. The understanding of how *h*BN interacts with
water can also provide insights to other processes that occur in aqu-
eous solutions, such as absorption, lubrication, catalysis, and
corrosion etc.

## Anti-scaling applications

Besides the flat Cu substrate, we grew *h*BN coating on both inside and
outside surfaces of stainless steel (SS) pipes using the chemical vapor
deposition (CVD) method (Fig. 5A), and assessed scale formation
behaviors of the pristine and *h*BN-coated SS pipes by pumping real
produced water collected from an oil & gas production site in Texas
through the pipes (Fig. 5B, Supplementary Note 5). These experiments
represent performance of the *h*BN coating on more realistic substrates
in complex solutions. As shown in Fig. 5C, significant scale formation
occurred inside the pristine SS tube after 7 days of operation, and its
effective inner diameter decreased by -17.2% after 14 days of operation,
resulting in a 31.3% reduction of cross-sectional area for water passage.
In contrast, the *h*BN-coated SS tube only experienced slight fouling by
organic matter in the produced water at the outlet (Fig. 5D), with no
measurable scale formation throughout the tube. These results
demonstrate the excellent anti-scaling potential of the *h*BN nano-
coating for real water and wastewater applications.

Overall, the unprecedented scaling resistance of *h*BN combined
with its other unique properties, including superior mechanical
strength[54,55], high dielectric constant[56], high thermal conductivity[57],
and chemical and thermal stability[58–61], make *h*BN an excellent candi-
date for multi-functional coatings in various industrial processes (e.g.
fluid transmission, heat exchange, membrane separation). These
coatings can protect substrate materials from scaling or corrosion[62], as
well as ensure energetically efficient fluid flow, high heat transfer rate,
and long material lifetime.

In summary, we discover that the unique combination of atom-
ically smooth surface, high in-plane polarity, and proper interatomic
spacing (lattice constant) makes *h*BN possibly the most anti-scaling
material known, with notable advantages over graphene. The atomic

smoothness of both $h$BN and graphene surfaces reduces the number of surface heterogeneous nucleation sites as well as the binding force between the mineral crystal and the surface, resulting in much greater resistance to mineral scaling compared to conventional scale-resistant materials. Interestingly, $h$BN exhibits even greater resistance to surface-induced heterogeneous nucleation than graphene, which stems from the in-plane dipole of the boron-nitrogen bond at a scale closely matching that of water molecules, leading to in-plane atomic energy corrugation that favors interaction with water. Direct experimental measurements and DFT calculations show that such chemical structure results in the formation of a dense hydration layer on the $h$BN surface, which strongly hinders mineral ions from approaching the surface and hence suppresses the heterogenous nucleation process. The $h$BN nanocoating grown on a stainless-steel tube exhibits outstanding anti-scaling properties in real oil & gas produced water, demonstrating its potential application in practical engineering systems. The results of the study provide important insights for future development of novel functional materials by manipulating their interactions with surrounding media. On the other hand, the scalability, durability, and long-term scaling resistance of the $h$BN coating as well as the specific roles of substrate materials and defects in large scale coatings need to be evaluated before practical applications can be possible.

## Methods

### Synthesis and characterization of the coatings

Ti, PVDF, graphene and $h$BN coatings were formed on Cu foil (25 μm in thickness, McMaster-CARR, USA). Before coating, the Cu substrate was electrochemically polished. Ti films of 40 nm in thickness were formed on the Cu substrate using a sputter coater (Denton Desk V, Denton Vacuum, USA) at a current of 30 mA. PVDF was coated on the Cu substrate via spin coating. A 1wt % PVDF in DMF solution was applied at a rotation speed of 2000 rpm for 1 min, followed by overnight drying at 60 °C before use.

Graphene and $h$BN films were formed on Cu substrate by chemical vapor deposition (CVD). During growth, the electrochemically polished Cu foil was first loaded into a tube furnace and heated to 1000 °C, followed by annealing at 1 Torr 15% $H_2$/Ar for 20 min. The graphene growth was then carried out by feeding 15% $H_2$/Ar (flow rate of 100 sccm) and methane (flow rate of 10 sccm) to the furnace for 20 min. For $h$BN growth, ammonia borane was used as the precursor. After Cu foil annealing, ammonia borane was heated to ~85 °C for evaporation upstream of the tube furnace, and was carried by $H_2$ gas to the Cu substrate to grow $h$BN at 1000 °C for 30 min. After growth with graphene or $h$BN, the Cu foil was rapidly cooled down to room temperature for further use.

After fabrication, the morphology of the four coatings were characterized by scanning electron microscopy (SEM, Quanta FEG 250, Thermo Fisher Scientific, USA) and atomic force microscopy (AFM) (NX20, Park, Suwon, Korea). The chemical property of the graphene and $h$BN were characterized by Raman spectroscopy (Renishaw inVia, UK) and X-ray photoelectron spectroscopy (XPS) (PHI Quantera II, Physical Electronics, USA) (Supplementary Note 1, Supplementary Figs. 1 and 2).

### Evaluation of scale formation on graphene and $h$BN using video camera

The graphene and $h$BN nanocoatings were formed on flat Cu substrates, and their scaling resistance was evaluated in a solution supersaturated with $CaSO_4$, a common scalant in ground water that was very difficult to remove. The supersaturated $CaSO_4$ solution (50 mmol $L^{-1}$) was prepared by mixing 100 mmol $L^{-1}$ $CaCl_2$ with 100 mmol $L^{-1}$ $Na_2SO_4$ at a 1:1 volume ratio. The prepared solution had a pH of 7.0 and a saturation index (SI) of 3.28. SI is defined as the ratio

between the chemical activity product of the mineral ions and their solubility product[63]. During the experiments, the pristine Cu foil, $h$BN- and graphene- grown Cu foils were immersed in the $CaSO_4$ solution. Scale formation was monitored using a video camera for 220 min (Supplementary Movie 1). The mass of mineral precipitation was determined by measuring the sample mass before and after the experiment.

### Ex situ measurement of scaling caused by heterogeneous nucleation on various coatings

Coated samples were cut into round wafers and installed in a flow cell. Supersaturated $CaCO_3$ solution was prepared by mixing $CaCl_2$ and $Na_2CO_3$ solutions, which were prepared using ultrapure water and aerated with air for 24 h to reach equilibrium before use. As shown in Fig. 2A, the equilibrated $CaCl_2$ and $Na_2CO_3$ solutions were continuously fed into an inline mixer at 0.1 mL $min^{-1}$ and then the flow cell at a combined flow rate of 0.2 mL $min^{-1}$. The chemistry of the mixed feed solution is simulated using Visual MINTEQ (version 3.1, KTH): $Ca^{2+}$ and $CO_3^{2-}$ activity are $1.23 \times 10^{-4}$ and $1.24 \times 10^{-4}$ mol $L^{-1}$, respectively; pH is 8.84, ionic strength is $5.56 \times 10^{-3}$, and calcite, aragonite, and vaterite saturation indexes ($\{Ca^{2+}\}\{CO_3^{2-}\}/K_{sp}$) are 4.45, 3.18, and 1.18 respectively. The induction time for homogeneous nucleation was determined by dynamic light scattering (DLS) measurements using a NanoBrook Omni (Brookhaven Instrument, Holtsville, NY, USA) (Supplementary Note 2, Supplementary Fig. 3). Since the hydraulic retention time from the mixer to the samples is much shorter than the induction time for $CaCO_3$ homogeneous nucleation, formation of $CaCO_3$ crystals on the sample surface is attributed to surface-induced heterogeneous nucleation. After the experiments, samples were retrieved and characterized using scanning electron microscopy (SEM, Quanta FEG 250, Thermo Fisher Scientific, USA) (Supplementary Note 2, Fig. 2B–E and Supplementary Fig. 4). The number and size of the crystals formed were analyzed using ImageJ software. More than 10 SEM images were analyzed for each surface to collect the crystal number and size data. The particle size distribution data were fitted with a Gaussian distribution function to obtain the average and standard deviation of particle size.

### Real time, in situ characterization of scale formation caused by heterogeneous nucleation using the QCMD technique

The heterogeneous nucleation and crystal formation of $CaCO_3$ was also characterized in situ and in real time using the quartz crystal microbalance with dissipation (QCMD) technique (Qsense E4 analyzer, Biolin Scientific, Sweden), which has a mass sensitivity and a dissipation sensitivity of 1.8 ng $cm^{-2}$ and $0.1 \times 10^{-6}$ in liquid, respectively[64]. To perform the QCMD experiments, the four test materials were first coated on QCMD sensors. Ti and PVDF were coated on QCMD sensors using the same methods as those used for coating the Cu substrate. Graphene and $h$BN were first grown on Cu substrate, and then transferred onto QCMD sensors using a PMMA-assisted method. In this method, a PMMA film was spin-coated on the graphene- or $h$BN- grown on the Cu foil. The sample was then immersed in a $FeCl_3$ solution to dissolve the Cu substrate. The graphene or $h$BN layer immobilized on the PMMA film was then transferred to DI water, and subsequently collected onto the QCMD sensor surface with the graphene or $h$BN side attaching to the sensor surface. Finally, acetone and IPA were used to dissolve the PMMA layer, exposing the graphene or $h$BN surface. The coated sensors were then mounted in 4 parallel QCMD cells, and characterized for $CaCO_3$ nucleation and growth at a temperature of 22 °C using the same supersaturated solution and flow rates as the offline scaling experiments described above. The frequency and dissipation data were continuously monitored for each coated sensor.

The increase of adsorbed or precipitated mass ($\Delta m$, ng $cm^{-2}$) on the sensors was calculated from the frequency change using the

Sauerbrey equation:

$$\Delta m = -C \cdot \frac{\Delta f}{n} \qquad (1)$$

Here C is the mass sensitivity constant, which is related to the properties of the quartz sensors and equals $17.7\,ng\,cm^{-2}\,Hz^{-1}$ in this study. $\Delta f$ is the frequency change (Hz). $n$ is the harmonic number.

## Binding force measurement

To measure the binding force between the vaterite crystals and the various surfaces that they grow from, samples were prepared by flowing the mixed $CaCl_2$ and $Na_2CO_3$ solution over the four surfaces for 15 h to allow vaterite growth. After crystal growth, the samples are transferred into a SEM assembled with nanoindenter equipment. To perform the measurement, the tip of the nanoindenter approaches and pushes a vaterite crystal grown on sample surface at a constant speed of $0.03\,\mu m\,s^{-1}$, until the vaterite crystal is detached from the underlying surface (Supplementary Note 3, Supplementary Fig. 6, Supplementary Movies 2 and 3). During this process, the applied load is recorded, and the applied load at the detaching point for the vaterite crystal is referred to as the detaching force. After normalizing the detaching force by contact area between the vaterite crystal and the surface, the obtained force reflects the binding force between the vaterite crystal and the underlying coating.

As shown in Supplementary Fig. 6 and Supplementary Movies 2 and 3, the nanotip can push the vaterite crystal at a direction perpendicular (Supplementary Fig. 6a, Supplementary Movie 2) or parallel (Supplementary Fig. 6b, Supplementary Movie 3) to the plane of the semi-hexagonal vaterite crystal. The detaching force differs greatly depending on the direction in which the load is applied (Supplementary Fig. 6a4, b4). This is due to the anisotropy structure of the vaterite crystal. As shown in Supplementary Fig. 6a, when pushing the crystal at the perpendicular direction, the detaching force is also greatly affected by the location of the contact point (e.g., distance from the coating surface). Therefore, data reported in the manuscript are all obtained by pushing the vaterite crystal in the direction parallel to the vaterite plane as that shown in Supplementary Fig. 6b.

## Simulation and calculation

In this study, we used density function theory (DFT) to evaluate (i) the charge distribution on graphene and $h$BN, (ii) the interaction between water molecules and $h$BN or graphene surface, (iii) the interaction of $Ca^{2+}$, $CO_3^{2-}$, and $CaCO_3$ with bare $h$BN and graphene, and (iv) the interaction between $Ca^{2+}$, $CO_3^{2-}$, and $CaCO_3$ and hydrated $h$BN or graphene. Detailed calculation and simulation methods can be found in the Supplementary Note 4 and Supplementary Figs. 7–11.

In addition to calculation, we also used the QCMD technique to quantitively verify the simulation results by measuring the adsorption of water vapor on graphene and $h$BN surfaces utilizing $N_2/H_2O$ mixed gas at different humidity levels. In these experiments, a customized humidity control device was installed between a $N_2$ gas tank and the QCMD analyzer (Fig. 4G). $N_2$ gas with different humidity was continuously flown into the QCMD analyzer at a flow rate of $7.5\,mL\,min^{-1}$. Corresponding frequency and dissipation change were recorded at 22 °C.

## Evaluating scale formation potential using real produced water

We also grew a $h$BN nanocoating inside a stainless-steel tube, and compared its scale forming behavior with an uncoated stainless-steel tube when exposed to a real oil & gas produced water taken from Texas. The produced water has a total organic carbon (TOC) concentration of $108\,mg\,L^{-1}$ and conductivity of 148 mS $cm^{-1}$. Detailed water quality data are shown in Supplementary Table 2 (Supplementary Note 5). As minerals have precipitated out from the solution during transportation of the produced water sample, the received produced water is no longer supersaturated (Supplementary Table 2) and cannot be directly utilized for scaling experiment. Instead, the oil & gas produced water is spiked with 50 mM $CaCl_2$ or $Na_2SO_4$ to prepare the respective $Ca^{2+}$- or $SO_4^{2-}$- rich solutions. During the experiment, the two solutions were continuously fed into an inline mixer at 0.1 mL $min^{-1}$ and then flowed through the pristine and $h$BN coated SS tubes at a combined flow rate of 0.2 mL $min^{-1}$. Scale formation was monitored using a camera during 14 days of operation.

## Data availability

All data generated or analyzed during this study are reported in this published article and its supplementary information files, and are available from the author upon request.

## Code availability

All codes supporting the findings of this study are available from the corresponding author upon request.

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

## Author contributions

All authors contributed intellectually to this paper. K.Z., X.H. designed and performed the experiments, analyzed data, and drafted the manuscript. Q.L., J.L., P.M.A., W.W., M.E. and P.J.J.A. conceived the idea, designed the experiments, revised the manuscript, and led the project. X.Z., T.Z., H.G. assisted sample growth and characterization. E.F.O. performed the simulation.

## Funding

This work was supported by the NSF Nanosystems Engineering Research Center for Nanotechnology-Enabled Water Treatment (EEC-1449500) (K.Z., X.H. and Q.L.), and the NSF I/UCRC Center for Atomically Thin Multifunctional Coatings (ATOMIC) under award # IIP-1539999 (T.Z., H.G. and J.L.). W.W. thank the financial support from NSFC under grant number 51788104 and 52001183. E.F.O. thank the Brazilian agencies CNPq and FAPESP (Grants 2013/08293-7, 2016/18499-0, and 2019/07157-9) for financial support, computational support from the Center for Computational Engineering and Sciences at Unicamp through the FAPESP/CEPID Grant No. 2013/08293-7 and the Center for Scientific Computing (NCC/GridUNESP) of São Paulo State University (UNESP).

## Competing interests

The authors declare no competing interests.
