## [Peer Review File · Nature Communications]

This manuscript has been previously reviewed at another journal that is not operating a transparent peer review scheme. This document only contains reviewer comments and rebuttal letters for versions considered at *Nature Communications*.

VIEWER COMMENTS

Reviewer #1 (Remarks to the Author):

This manuscript by Zuo and co-authors reports on the excellent scaling resistance (resistance to surface mineral growth or deposition from super-saturated solutions) of hBN materials. The article reports a large number of experimental results, from macroscopic video observation, quartz-crystal microbalance (QCM) measurements, adhesive/binding forces of crystals to the surface, water adsorption measurement by QCM and macroscopic scale mitigation through hBN nano-coatings. On the computational side, the authors also present an extensive investigation of the interaction of hBN and graphene surfaces with water molecules and solvated ions (which would be precursors to grown crystals). The main finding of the paper is the large scaling resistance of hBN, compared to other standard surfaces, even when compared with graphene. Based on molecular dynamics simulations, the authors provide a tentative interpretation, related to hBN atomic smoothness, as well as from its polar nature, which would lead to strong affinity to water molecules, preventing surface crystal nucleation. Investigation and report of this particular scaling property of hBN substrate appears original, and its relation to molecular surface properties of the material is interesting. The experimental and computational work brings together different scales, from macroscopic, real-life investigation down interfacial properties at the molecular scale. This brings an interesting perspective on the topic. As this manuscript contains a number of novel interesting observations and appears technically sound, I would recommend its publication in Nature Communication following clarification of the following points: 1. Regarding the QCM measurements presented on page 8, the authors state that “On the hBN surface, the frequency and dissipation shifts throughout the experiment were very small, indicating no measurable crystal growth during this period.” This is not the case as a clearly measurable frequency shift indeed appears on the QCM measurements (Fig. 2D, red and green). The signal is indeed smaller than for PVDF and Ti but is clearly not zero or noise-limited. The author should temper their claims and explain why is that? Regarding this QCM measurement, would it also be possible to extract a quantitative value for the added mass per electrode area. It would provide much interesting and richer information than only the frequency shift, that the authors could compare to their other scale formation measurements. 2. The authors claim that surface roughness is key to hBN and graphene scaling resistance. As both 2D materials are deposited on very smooth silicon wafers, one may wonder what is the scaling resistance of the bare wafer and how does it compare to the ones coated with 2D materials? 3. The point regarding “ultra-fast liquid transport”, mentioned on page 10 is unclear. First, friction is related to dynamic flow, not clearly related to scaling. Second, some studies have shown that hBN and graphene apparently possess very different friction properties (high friction of hBN and low friction on graphene). This material property should be put in the context of the study (see point 8 below). Also the citations associated with the mention on “ultra-fast transport” do not all refer to liquid flow. The authors could potentially replace by a statement more general like “peculiar interactions of 2D materials with water”. 4. Defects in the crystals (point defects or line defects) could also play an important role for the nucleation. Could the authors comment on that? 5. The energy reported in Fig. 4D seems unrealistically large. If it is already of order 20 eV on graphene, then Ca²⁺ adsorption should never occur and the additional 18 eV compared to hBN do not really matter.. This is surprising and should be clarified. 6. The authors should at also discuss surface charging of the two materials, which could also be important in this process. In the experiment, the role of pH might then be important. 7. I am unsure about the interpretation concerning the comparison of the interatomic spacing with the size of water molecule for hBN both in the text and in the abstract. Indeed, the difference between hBN and graphene appears very small. Also for disordered liquids versus solid surfaces, would this

corrugation effect really matter for these interfacial properties 8. Regarding the dissipation measurement of the QCM in humid air, the result is interesting. But the author should explain more clearly what the ratio dissipation/frequency shift means (I guess it is the associated dissipation normalized by water thickness). The terms “compact and rigid” and “softer and viscous” are not very rigorous. I guess slippage at the substrate/water interface should also affect these dissipation measurements. For the discussion regarding these points, the authors could also point toward the references they cite above regarding fast (and slow) liquid transport over carbon (versus hBN). Minor points: - I did not know the term “scale formation” before. Maybe the author could also introduce it with more physically oriented terms, talking e.g. of crystal nucleation/deposition. - In Figure 4, as the author can extract a thickness from Δf , they should show it, as it carries more meaning than Δf . At least in E3.

Reviewer #2 (Remarks to the Author):

This is a well put together paper on the ability of hexagonal boron nitride to resist mineral scale. The authors have presented both experimental and modeling results to demonstrate the characteristics of hBN that make it attractive as an anti-scale coating.

I found the paper to be complete in its examination of the coatings. There were only a few things I think need to be addressed:

1. Figure 2D - You say that the frequency returns to baseline, but it seems there is still a shift for everything except graphene. Can you comment on why hBN does not quite return to the baseline level, even though the frequency shift is minor?
2. Figure quality - in many figures, the text is too small to read (scales on AFM images in Fig 3 for example). Can you please increase text size to be readable.
3. In the QCMD experiments, you removed the scale easily by flushing solution through the system. In the real water situation though, you did not attempt to remove scale. Did you notice that the scale in these pipe systems was easily removable? Could flushing remove the observed scale in these cases?
4. Figure S1 refers to a "similar transfer method." Could you please clarify what you mean here? I saw no reference to a transfer method in the text. The images used here also seem to be fairly poor quality.
5. The longevity of coatings would also be of interest. Can you comment on the ability of these coatings to 1) be cleaned, 2) resist over time and 3) the general durability on surfaces?

Overall, this is a nice manuscript.

Reviewer #3 (Remarks to the Author):

Formation of mineral scale on a material surface has been a big problem for a variety of applications. Understanding and controlling mineral scaling behaviour is very important for R&D of next generation materials and technologies. This ms reports the superior resistance of hexagonal boron nitride (hBN) to mineral scale formation compared to not only common metal and polymer surfaces but also to graphene, which is the highly resistant to scaling, making hBN the most robust and scaling resistant material. It is a very interesting idea and an important study in the field of 2D nanomaterials.

In this ms, the detailed experimental and simulation results showed that this ultrahigh scaling-resistance is attributed to the combination of hBN's atomically-smooth surface, in-plane atomic energy corrugation due to the polar boron-nitrogen bond, and interatomic spacing that matches

closely with the size of a water molecule. Furthermore, atomically hBN are normally considered to be chemically inert, which will makes hBN successful and useful in the surface coating industry.

I would recommend this ms is accepted for a publication in Nature Communications after addressing the following comments.

The specific comments are:

1. In evaluation of scale formation on graphene and hBN, a supersaturated CaSO₄ solution was used, what is pH value of this solution? pH would play a role for the scale formation?
2. In method, saturation index was given, what does it mean? The brief introduction and refs are required.
3. The QCMD technique was used to real-time characterise scale formation, what is the sensitivity of QCMD sensor? I suggest some references should be added.

REVIEWER COMMENTS

Reviewer #1 (Remarks to the Author):

This manuscript by Zuo and co-authors reports on the excellent scaling resistance (resistance to surface mineral growth or deposition from super-saturated solutions) of hBN materials. The article reports a large number of experimental results, from macroscopic video observation, quartz-crystal microbalance (QCM) measurements, adhesive/binding forces of crystals to the surface, water adsorption measurement by QCM and macroscopic scale mitigation through hBN nano-coatings. On the computational side, the authors also present an extensive investigation of the interaction of hBN and graphene surfaces with water molecules and solvated ions (which would be precursors to grown crystals). The main finding of the paper is the large scaling resistance of hBN, compared to other standard surfaces, even when compared with graphene. Based on molecular dynamics simulations, the authors provide a tentative interpretation, related to hBN atomic smoothness, as well as from its polar nature, which would lead to strong affinity to water molecules, preventing surface crystal nucleation.

Investigation and report of this particular scaling property of hBN substrate appears original, and its relation to molecular surface properties of the material is interesting. The experimental and computational work brings together different scales, from macroscopic, real-life investigation down interfacial properties at the molecular scale. This brings an interesting perspective on the topic.

As this manuscript contains a number of novel interesting observations and appears technically sound, I would recommend its publication in Nature Communication following clarification of the following points:

Response: The authors thank the reviewer for the helpful comments and suggestions, which have greatly improved the quality of our article. The manuscript has been revised accordingly, and the point-to-point responses are also provided as below.

1. Regarding the QCM measurements presented on page 8, the authors state that “On the hBN surface, the frequency and dissipation shifts throughout the experiment were very small, indicating no measurable crystal growth during this period.” This is not the case as a clearly measurable frequency shift indeed appears on the QCM measurements (Fig. 2D, red and green). The signal is indeed smaller than for PVDF and Ti but is clearly not zero or noise-limited. The author should temper their claims and explain why is that? Regarding this QCM measurement, would it also be possible to extract a quantitative value for the added mass per electrode area. It would provide much interesting and richer information than only the frequency shift, that the authors could compare to their other scale formation measurements.

Response: The authors thank the reviewer for this comment. The reviewer is correct that the frequency and dissipation shifts were very small but measurable. We did not overlook these measurable shifts. The mineral crystal formation process consists of two steps: nucleation and crystal growth. The nucleation step, i.e., induction, involves a very small amount of aggregated mass due to the small size of the nuclei. The mass of the solid phase increases much faster in the crystal growth step, when the mineral precipitates on the previously formed nuclei. As shown in the data for graphene, PVDF and Ti surfaces, the changes of frequency (dF/dt) and dissipation (dD/dt) in the QCMD experiments can be

used to detect the occurrence of nucleation and crystal growth as the data exhibit two distinct stages of different frequency and dissipation shift rates. In the first stage, changes in frequency and dissipation signals are small and slow, reflecting the very small mass of nuclei formed on the surface. The duration of this stage is therefore considered the induction time for crystal formation. In the second stage, there is a clear, large increase in the frequency and dissipation shift rates; this signals the formation and growth of crystals due to precipitation of CaCO₃ on the previously formed nuclei. Both stages were observed on Ti, PVDF, and graphene surfaces, suggesting that crystals were grown on these surfaces. On the hBN surface, however, the frequency and dissipation shifts throughout the experiment were very small, and there was not an observable sudden change in the frequency and dissipation shift rate, i.e., it did not exhibit an obvious second stage. This suggests that the length of the experiment was shorter than the induction time on the hBN surface, and crystal growth did not occur during the duration of the experiment. We have revised the text to better clarify the interpretation of the QCMD data. Please see line 153 – 166 in the revised manuscript.

“This was followed by a much faster frequency shift (stage ii), which signaled crystal growth from the nuclei previously formed on the surface (Fig. 2D2). CaCO₃ mass accumulation rate calculated using the Sauerbrey equation (Fig. S5) showed a very large increase in stage ii compared to that in stage i on Ti, PVDF and graphene surfaces. As shown in Fig. 2D2 & D3, the Ti surface showed the shortest induction time and the fastest crystal growth kinetics, followed by PVDF. The heterogeneous nucleation induction time on graphene was significantly longer compared to Ti and PVDF (Fig. 2D2), and a much slower crystal growth kinetics was observed (Fig. 2D3 and S5). On the hBN surface, the frequency and dissipation shifts throughout the experiment were very small. It did not exhibit a distinct stage ii with fast frequency and dissipation changes. This suggests that the induction time on the hBN surface is longer than the approximately 8 h contact time with the test solution in the experiment; no detectable crystal growth occurred during this period (Fig. 2D2, D3 & Fig. S5). This may be attributed to the low ion adsorption on hBN (Fig. 2D2), which prolongs the induction of heterogeneous nucleation and hinders vaterite crystal growth.”

Based on the suggestion from the reviewer, we calculated the mass accumulation rate on various coatings from the QCMD data. This was done using the Sauerbrey equation. We described the method of the calculation in the method section of manuscript, reported detailed calculation results in the supporting information, and discussed the results in the main text. The revisions made include the following:

Line 429 – 434 in the revised manuscript:

“The increase of adsorbed or precipitated mass (Δm , ng cm⁻²) on the sensors was calculated from the frequency change using the Sauerbrey equation:

$$\Delta m = -C \cdot \frac{\Delta f}{n} \quad (1)$$

Here C is the mass sensitivity constant, which is related to the properties of the quartz sensors and equals 17.7 ng cm⁻² Hz⁻¹ in this study. Δf is the frequency change (Hz). n is the harmonic number.”

Fig. S5. Mass increase rates on various surfaces during the nucleation and crystal growth stages of CaCO₃ scale formation.”

2. The authors claim that surface roughness is key to hBN and graphene scaling resistance. As both 2D materials are deposited on very smooth silicon wafers, one may wonder what is the scaling resistance of the bare wafer and how does it compare to the ones coated with 2D materials?

Response: In our experiments, graphene and hBN were grown on Cu substrate, not silicon wafers. Silicon wafers have very smooth and hydrophilic surface, and are known to have high scaling resistance. However, it cannot be used as a coating material. Therefore, it is not included in our study.

The reviewer raises an excellent point on the effect of substrate, especially for ultra-smooth surfaces such as graphene and hBN. The roughness of the substrate may impact the roughness of the thin 2-D material coating and therefore the scale formation kinetics. In our QCMD study, hBN and graphene were transferred to gold (Au)- coated QCMD quartz sensors (roughness < 1 nm). We have conducted additional experiments to evaluate the scaling resistance of the bare Au-coated quartz sensor. As shown in the figure below, the hydrophilic, smooth Au-coated quartz sensor (Au@SiO₂) exhibited very small frequency changes similar to that of hBN. Therefore, the impact of the substrate is considered negligible. Although the impact of substrate surface roughness was not systematically investigated, our experiments using a stainless-steel pipe show that the hBN nanocoating retains its anti-scaling properties on realistic, rougher substrates. At the same time, we recognize the need to more systematically understand the impact of substrate properties. We have added text to clarify these points.

Line 318 – 319 in the revised manuscript:

“These experiments represent performance of the hBN coating on more realistic substrates in complex solutions.”

Lines 355 – 358 in the revised manuscript:

“On the other hand, the scalability, durability, and long-term scaling resistance of the hBN coating as well as the specific roles of substrate materials and defects in large scale coatings need to be evaluated before practical applications can be possible.”

Figure. Real time *in-situ* monitoring of CaCO₃ scale formation on Ti, PVDF, graphene, hBN, and Au@SiO₂ using QCMD.

3. The point regarding “ultra-fast liquid transport”, mentioned on page 10 is unclear. First, friction is related to dynamic flow, not clearly related to scaling. Second, some studies have shown that hBN and graphene apparently possess very different friction properties (high friction of hBN and low friction on graphene). This material property should be put in the context of the study (see point 8 below). Also the citations associated with the mention on “ultra-fast transport” do not all refer to liquid flow. The authors could potentially replace by a statement more general like “peculiar interactions of 2D materials with water”.

Response: The authors thank the reviewer for this helpful comment. We agree with the reviewer and have revised this sentence as suggested by the reviewer. Please see line 211 – 215 of the revised manuscript.

“The lower binding force on the atomically-smooth graphene and hBN surfaces measured in our study is consistent with findings from previous studies that show atomically smooth carbon and boron-nitride materials exhibit super-lubricity at solid/solid rigid junctions^{35, 36, 37, 38} and peculiar interactions with water in 1D nanotubes^{22, 39, 40, 41} or 2D structures^{21, 23, 24, 42, 43, 44} at their solid/liquid interfaces.”

In addition, we added discussion and a reference in the revised manuscript to explain that hBN has stronger water adsorption and higher water friction on its surface than graphene. Please see line 249 – 252 in the revised manuscript:

“Similar results have been reported in previous studies. Utilizing ab initio molecular dynamics simulation, it was reported that the friction coefficient on hBN was about 3 times larger than that on graphene, and the adsorbed water molecule had faster slippage on graphene than on hBN¹⁸.”

Line 539 – 541 in the revised manuscript:

“18. Tocci, G., Joly, L., Michaelides, A. Friction of water on graphene and hexagonal boron nitride from ab initio methods: very different slippage despite very similar interface structures. Nano Lett. 14, 6872-6877 (2014).”

4. Defects in the crystals (point defects or line defects) could also play an important role for the nucleation. Could the authors comment on that?

Response: The authors agree with the reviewer for the importance of the *h*BN coating quality. Defects create nucleation sites for scale formation. In our study, SEM images showed that the *h*BN and graphene samples prepared had high quality on the millimeter-scale Cu substrate. No defects were detected in any samples we characterized using SEM. Line defects can usually be identified from the patterns of the mineral nuclei or crystals formed. As shown in the images in Figures 1 and 2, no obvious signs of line defects were observed. Small point defects (e.g., vacancies) are more difficult to detect. If they are present, they are very likely nucleation sites where mineral crystallization starts. Very importantly, it is difficult to avoid defects in fabrication of large-scale *h*BN and graphene coatings in real applications. Therefore, we have added discussion in the conclusion section of the revised manuscript to emphasize this point. Please see line 355 – 358 in the revised manuscript:

*“On the other hand, the scalability, durability, and long-term scaling resistance of the *h*BN coating as well as the specific roles of substrate materials and defects in large scale coatings need to be evaluated before practical applications can be possible.”*

5. The energy reported in Fig. 4D seems unrealistically large. If it is already of order 20 eV on graphene, then Ca²⁺ adsorption should never occur and the additional 18 eV compared to *h*BN do not really matter. This is surprising and should be clarified.

Response: We thank the reviewer for this comment. These energy number did seem unrealistic large, this was caused by the limitation of the calculation method previously used.

The simulation presented in our original manuscript (Fig. 4D) estimates the necessary energy for a Ca²⁺ to diffuse through the closest water layer in contact with *h*BN and graphene. The simulation was designed for comparison purposes, and not meant to quantify the absolute value of the energy barrier. To do such analysis, we placed a Ca²⁺ initially ~3.0 Å away from the closest water layer that was in contact with *h*BN and graphene; then, we created several (frozen) atomic conformations in which we moved manually the Ca²⁺ position from ~3.0 Å from the water layer until it reached the other side, ~6.0 Å distant from the initial position, with a step of 0.5 Å. If we consider that the water layer was at position 0.0 Å, the variation of the position of Ca²⁺ was from ~-3.0 Å to ~3.0 Å. In other words, we created manually a trajectory for Ca²⁺ diffusion through the water layer. For each configuration, we performed an energy measurement and built the Fig. 4D. However, this methodology does not explore all possible paths for Ca²⁺ to cross the water layer, neither does it seek the path of the lowest energy barrier. As a result, the paths simulated had very high energy barrier.

To address this issue, we have redone the simulation with a more reliable method that can find minimum energy paths between the defined initial and final states, called Nudge Elastic Band (NEB). This method works by optimizing several intermediate images (each with a specific atomic configuration) between the initial and final states and creating

a reaction path. Each image finds the lowest energy possible path to follow while maintaining equal spacing to neighboring images. This allows finding the most energy-efficient path among the potential paths simulated. Results of the revised simulation are shown in the revised Figure 4D. The revised simulations show that Ca^{2+} needs to overcome an energy barrier of 3.3 and 2.3 eV to access the *h*BN and graphene surface, respectively. An additional 1.0 eV is necessary to cross the closest hydration layer on the *h*BN surface compared with that on graphene surface. These values of the energy barrier and energy barrier difference may still be not accurate because the water molecules in the hydration layer on the *h*BN and graphene surfaces are treated as frozen (not mobile) in the simulations. This may lead to stronger interactions between the Ca^{2+} ion and water molecules, i.e., the calculated energy barrier may be higher than the actual values. However, the qualitative trend should remain the same: it is more difficult for Ca^{2+} to reach the *h*BN surface than the graphene surface. The difference in energy barrier clearly reflects the difference in the structure of the water layer on *h*BN versus graphene surfaces. We have revised the text in the main manuscript as well as the supporting information.

Figure 4D Minimum energy needed for a Ca^{2+} ion to penetrate the hydration layer on *h*BN and graphene surface;”

Line 260 – 262 in the revised manuscript:

“More energy (1.0 eV) is required for a Ca^{2+} ion to penetrate the hydration layer to reach the *h*BN surface than the graphene surface (Fig. 4D).”

Page 12 in the revised supporting information:

“In order to test this hypothesis, we performed an exploratory simulation using the Nudge Elastic Band (NEB) method⁸ to determine the minimum energy for a Ca^{2+} ion to move through the closest layer of water molecules on *h*BN and graphene surfaces. The NEB method looks for a favorable path with minimum energy requirement for the Ca^{2+} ion to penetrate the water layer. As shown in Figure 4D in the manuscript, the necessary energy increases significantly with the approaching of Ca^{2+} to the water layer; 1.0 eV more energy is needed for Ca^{2+} to penetrate the water layer on *h*BN than on graphene. The absolute values of the energy barrier and energy barrier difference may not be accurate because the hydration water molecules on the *h*BN and graphene surfaces are set as “frozen”, i.e., immobile, in the simulation. This may result in higher calculated energy barriers. However,

the qualitative trend should remain the same: it is more difficult for Ca^{2+} to reach the hBN surface than the graphene surface.”

Page 14 in the revised supporting information:

“8 Jonsson, H., Mills, G. & Jacobsen, K. W. in Classical and Quantum Dynamics in Condensed Phase Simulations 385-404.”

6. The authors should at also discuss surface charging of the two materials, which could also be important in this process. In the experiment, the role of pH might then be important. **Response:** Surface charge plays an important role in mineral scale formation. The surface charge is often characterized by zeta potential, and it is a function of solution pH. The zeta potential of graphene and hBN have been reported in previous studies, which are shown in the figure below.

Figure. Zeta potential of graphene and hBN at various solution pH. The hBN data was reproduced from an open access article³. The graphene data was reproduced with permission of copyright © 2008 Nature Publishing Group⁴.

As shown in the figure above, the zeta potentials of graphene and hBN are very similar within the pH range of 3.0-11.0, and have similar point of zero charge (PZC) at ~ pH 3.3. In our study, the CaCO_3 solution had a pH of 8.84. Therefore, graphene and hBN are expected to have similar surface charge. This is consistent with the similar ion adsorption measured on both surfaces (Figure 2D1).

Solution pH affects not only material surface charge (zeta potential), but also the precipitation reaction equilibrium. Speciation of carbonate in water is a strong function of pH. Therefore, any changes in pH will not only change material surface charge, but also the saturation index for a given solution. Therefore, we kept pH constant in all experiments. We have added discussion in the revised manuscript to comment the effect of surface charge on graphene and hBN, please see line 148 – 150 in the revised manuscript:

“The similar frequency and dissipation changes on graphene and hBN surface suggested that they had similar ion adsorption, a result of their similar surface charge^{31, 32}.”

Line 585 – 590 in the revised manuscript:

“31. Qu, J., Li, Q., Luo, C., Cheng, J., Hou, X. Characterization of flake boron nitride prepared from the low temperature combustion synthesized precursor and its application for dye adsorption. Coatings 8, 214 (2018).”

32. *Li, D., Müller, M. B., Gilje, S., Kaner, R. B., Wallace, G. G. Processable aqueous dispersions of graphene nanosheets. Nat. Nanotechnol. 3, 101-105 (2008)."*

7. I am unsure about the interpretation concerning the comparison of the interatomic spacing with the size of water molecule for hBN both in the text and in the abstract. Indeed, the difference between hBN and graphene appears very small. Also for disordered liquids versus solid surfaces, would this corrugation effect really matter for these interfacial properties.

Response: The authors thank the reviewer for this comment. We agree with the reviewer that the difference of interatomic spacing (lattice constant) between the hBN (2.50 Å) and graphene (2.46 Å) is small. Both are significantly smaller than the lattice constants of other materials, e.g., NaCl, Au, Cu, and match better with the water molecule size. The small difference between the lattice constants of hBN and graphene is not our explanation of the difference in water layer structure observed on hBN and graphene surfaces. The main reason for the observed difference in hydration water structure is the charge corrugation on hBN surface (due to the polarity of the B-N bond) versus the more uniform charge distribution on graphene. The close match between the hBN lattice constant and the water molecular size leads to the alignment of the hBN surface charge corrugation (i.e., in-plane polarity) with the dipole of water molecules. As a result, the density of the water layer on the hBN surface is higher than that on the graphene surface.

Regarding the second comment, the authors agree with the reviewer that the surface charge corrugation may not have an important impact for a disordered liquid phase. However, unlike those in the bulk phase, water molecules in the hydration layer on a solid surface is highly ordered,^{5, 6, 7} which can also be seen in the DFT calculations in our study. The charge corrugation on the hBN surface has an important effect on how the hydration water molecules are packed on the surface, i.e., the structure of the ordered hydration layer, which in turn impact the nucleation of scale-forming minerals.

We have revised the abstract and the main text to better elucidate this analysis. Detailed revision can be seen below:

Line 36 – 41 in the revised abstract:

"Experimental and simulation results reveal that this ultrahigh scaling-resistance is attributed to the combination of hBN's atomically-smooth surface, in-plane atomic energy corrugation due to the polar boron-nitrogen bond, and the close match between its interatomic spacing and the size of water molecules. The latter two properties lead to strong polar interactions with water and hence the formation of a dense hydration layer, which strongly hinders the approach of mineral ions and crystals, decreasing both surface heterogeneous nucleation and crystal attachment."

Line 293 – 296 in the revised manuscript:

"The close match between the distribution of hBN surface energy corrugation (charge nonuniformity) and the di-pole of water molecules leads to strong polar interactions between the hBN surface and polar water molecules, and hence compact packing of water molecules on hBN surface."

8. Regarding the dissipation measurement of the QCM in humid air, the result is interesting. But the author should explain more clearly what the ratio dissipation/frequency shift means (I guess it is the associated dissipation normalized by water thickness). The terms “compact and rigid” and “softer and viscous” are not very rigorous. I guess slippage at the substrate/water interface should also affect these dissipation measurements. For the discussion regarding these points, the authors could also point toward the references they cite above regarding fast (and slow) liquid transport over carbon (versus hBN).

Response: The authors thank the reviewer for the valuable comment. The term dissipation (D) refers to is the energy loss of the quartz crystal sensor resulting from the interaction of the adsorbed layer with the surrounding fluid⁸, and is measured by the decrease in quartz crystal sensor vibration amplitude when the applied voltage is turned off. Therefore, the ratio of dissipation change (ΔD) to frequency change (Δf) (Df ratio) is directly associated with the viscoelastic properties of the adsorbed/deposited layer. We agree with the reviewer that the term “softer and viscous” is not very rigorous, and have removed these ambiguous descriptors from the manuscript. The text in the manuscript has been revised to explain more clearly the physical meaning of the dissipation shift/frequency shift ratio.

We also agree with the reviewer that slippage can affect dissipation, which has been observed in both gaseous and liquid phases^{9, 10, 11}. In this study, the Df ratio of the hydration layers on graphene is larger than that on hBN surfaces. This can be caused by two possible differences, a less compact hydration layer and/or water slippage. Prior research has shown that the contribution of slippage to energy dissipation shift for a 5 MHz sensor is very small: less than 1%^{9, 10, 11}. Therefore, we attribute the higher Df ratio observed on graphene to the less compact structure of the hydration layer formed on graphene surface.

Please see line 271 – 273 in the revised manuscript:

“As water slippage on surfaces has little impact on ΔD and ΔF ^{46, 47}, the measured $\Delta D/\Delta F$ directly reflects the viscoelastic properties of the adsorbed water layer⁴⁸.”

Line 638 – 647 in the revised manuscript:

*“46. Huang, K., Szlufarska, I. Friction and slip at the solid/liquid interface in vibrational systems. *Langmuir* 28, 17302-17312 (2012).*

*47. Johannsmann, D., Reviakine, I., Richter, R. P. Dissipation in films of adsorbed nanospheres studied by quartz crystal microbalance (QCM). *Anal. Chem.* 81, 8167-8176 (2009).*

*48. Cho, N.-J., Frank, C. W., Kasemo, B., Höök, F. Quartz crystal microbalance with dissipation monitoring of supported lipid bilayers on various substrates. *Nat. Protoc.* 5, 1096-1106 (2010).”*

Minor points:

9. I did not know the term “scale formation” before. Maybe the author could also introduce it with more physically oriented terms, talking e.g. of crystal nucleation/deposition.

Response: We thank the review for the suggestion. We have revised the text in the introduction section to include a brief definition of scale formation in line 43 – 46 in the revised manuscript.

“In particular, scale formation, i.e., the development of mineral deposits on a material surface due to precipitation from the bulk solution and/or crystal formation initiated by

surface nucleation, has great impacts on interfacial transfer of mass, heat, electrons, and light.”

10. In Figure 4, as the author can extract a thickness from Δf , they should show it, as it carries more meaning than Δf . At least in E3.

Response: The authors agree with the reviewer that thickness of the hydration layer on different coating surfaces, if can be obtained, would be a more direct and visual measure of the adsorbed water layer. However, the determination of the adsorbed layer thickness data requires the density of the adsorbed layer. As shown in our DFT simulation and previous studies, the structure and density of the hydration layer on *h*BN and graphene surfaces are different. Therefore, hydration layer thickness calculated can only be used as a reference value to show difference in the hydration layer. To avoid misleading the readers, we decided to remove Fig. S11 and the related text.

As a comparison, ΔF is a direct experimental measurement and does not involve any assumptions. Difference in ΔF reflects difference in adsorbed mass, which can present itself as difference in density and/or thickness. In addition, the lower $\Delta D/\Delta F$ on *h*BN than that on graphene shown in Figure 4E3 reflects the higher density of the adsorbed water layer on *h*BN surface.

Based on the considerations discussed above, we believe it is better to keep the ΔF and $\Delta D/\Delta F$ data in Figure 4E3, and remove Fig. S11 and related discussion on water layer thickness from the supporting information. Revisions in the text in the manuscript and supporting information are listed below.

A sentence has been removed from line 257 – 258 in the original manuscript:

*“the adsorbed water film is also thicker on *h*BN (0.03–2.89 nm) than that on graphene surface (0.01–1.68 nm, Fig. S11).”*

Some text has been removed from line 448 – 457 in the original manuscript:

*“The thickness (t , nm) of water film formed on the *h*BN and graphene surface was calculated using Sauerbrey equation:*

$$t = C \frac{\Delta f}{n \cdot d} \quad (1)$$

Here C is the mass sensitivity constant, it is related to the properties of the quartz sensors and equals $17.7 \text{ ng cm}^{-2} \text{ Hz}^{-1}$ in this study. Δf is the frequency change (Hz); n is the number of the harmonic; d is the density of water layer formed. We use 1 g cm^{-3} for d to simplify the calculation despite the difference in d for different material surfaces. A higher t would therefore mean a thicker and/or denser water layer. The calculated thickness values are shown in Fig. S11 in the Supplementary section 4.”

Fig. S11 and the related text in the original supporting information has been removed:

*“In addition to the DFT and MD calculations, experiments were also performed using the QCMD technique to quantitatively measure water adsorption on *h*BN and graphene surfaces. Utilizing N_2 gas with different humidity, water adsorption on *h*BN or graphene is characterized by the frequency change as shown in Fig. 4E in the manuscript. The frequency change increases with the influent N_2 gas humidity, indicating increased water adsorption with high influent humidity. If assuming the density of water film on *h*BN and*

graphene surface are both 1 g cm^{-3} , the thickness of water film on hBN and graphene surfaces are shown Fig. S11. It is noted that the density of the water film formed on hBN and graphene surfaces are most likely different as shown in Fig. S7. Therefore, a larger thickness calculated here can mean a thicker and/or denser water film. As shown in Fig. S11, the calculated water film thickness on hBN is always greater than that on graphene, indicating adsorption of more water molecules on hBN than that on graphene.

Fig. S11. Thickness of water film formed on the hBN and graphene coated QCMD sensor at different feed gas humidity.”

Reviewer #2 (Remarks to the Author):

This is a well put together paper on the ability of hexagonal boron nitride to resist mineral scale. The authors have presented both experimental and modeling results to demonstrate the characteristics of hBN that make it attractive as an anti-scale coating.

I found the paper to be complete in its examination of the coatings. There were only a few things I think need to be addressed:

1. Figure 2D - You say that the frequency returns to baseline, but it seems there is still a shift for everything except graphene. Can you comment on why hBN does not quite return to the baseline level, even though the frequency shift is minor?

Response: The authors thank the reviewer for this comment. The reviewer is correct that the frequency shift did not return to zero except for graphene. We attribute this to the incomplete dissolution of the nuclei and crystals and incomplete desorption of the adsorbed ions on the Ti, PVDF, and hBN surfaces within the limited time of DI rinse. It can be seen in Figure 2D2 that the frequency shift curve of hBN still has a significant positive slope, suggesting that Δf is still moving towards zero. The Δf value at the end of the experiment (~14 h) was very close to that at 2 h when Ca^{2+} and CO_3^{2-} ions were initially adsorbed on the hBN surface, suggesting that there may still be adsorbed ions on the surface. All surfaces used in the experiment were examined using SEM after the experiment, and no CaCO_3 crystals were observed on any surfaces. Therefore, we can attribute the very small amount of remaining frequency shift to the remaining adsorbed ions. We have revised this sentence in line 166 – 170 of the revised manuscript to note and explain the residual frequency shift:

“After 10 h of operation, the system was flushed with ultrapure water. Frequency and dissipation signals approached the original DI water baseline at different rate for all

samples (Fig. 2D2), confirming that the frequency and dissipation changes observed earlier resulted from the nucleation and growth of vaterite, which was removed or dissolved when flushed with ultrapure water.”

2. Figure quality - in many figures, the text is too small to read (scales on AFM images in Fig 3 for example). Can you please increase text size to be readable.

Response: The authors thank the reviewer for this suggestion. Font size in the AFM images in Figure 3, the inserted plots in Figure 2B1-B4 has been increased for better readability. The revised Figure 3 and Figure 2 are shown below. We also made sure that text in other figures are readable.

Figure 3 in the revised manuscript:

Figure 3. Atomically-smooth morphology yields low scalant binding force. (A) AFM image of (A1) Ti, (A2) PVDF, (A3) graphene, (A4) hBN surfaces; **(B)** Measurement of vaterite detaching force from various coatings. (B1) Sectional view schematic showing the measurement method by adding a lateral force on a vaterite crystal grown on a sample surface. Top view SEM image of a vaterite crystal (B2) before and (B3) after being pushed off from the sample surface by the nanoindenter tip. The scale bars in B2 and B3 represent 10 μm. The dashed circle in B3 shows the contact area between the vaterite crystal and the underlying sample surface after detachment; (B4) Applied force load increases linearly with the moving depth of the nanotip upon contact to the vaterite crystal; (C) The average vaterite detaching force and surface roughness of the four nanocoatings. Error bars represent the standard deviation of corresponding results.”

Figure 2 in the revised manuscript:

Figure 2. Characterization of surface-induced heterogeneous nucleation of CaCO_3 on Ti, PVDF, graphene, and hBN. (A) A schematic of QCMD experimental setup for characterizing CaCO_3 heterogeneous nucleation on four surfaces. SI: saturation index, PC: personal computer, QCMD: quartz crystal microbalance with dissipation; (B) SEM images of CaCO_3 crystals formed on (B1) Ti, (B2) PVDF, (B3) graphene, and (B4) hBN coatings on Cu substrate after 9 h of operation, with insets showing the size distribution of vaterite crystals formed. The particle size distribution shown in the B1-B4 inserts are obtained using the ImageJ software and fitted with a Gaussian distribution function; (C) Average number and size of vaterite crystals grown on the four surfaces (C1) as a function of operation time and (C2) after 15 h of operation; (D) Real time in-situ monitoring of CaCO_3 scale formation using QCMD. (D1) Ion adsorption. (D2) Induction and growth of CaCO_3 crystals. Three arrows marked on the Ti, PVDF, and graphene curves indicate the start and finish of two stages: the induction stage i and the growth stage ii. Data obtained using the hBN sample only exhibit one stage (stage i) during the operation. (D3) Average frequency change rate ($\Delta F/\Delta t$) during the two-stage scale-formation process and surface roughness of the four nanocoatings. Error bars represent the standard deviation. ”

3. In the QCMD experiments, you removed the scale easily by flushing solution through the system. In the real water situation though, you did not attempt to remove scale. Did you notice that the scale in these pipe systems was easily removable? Could flushing remove the observed scale in these cases?

Response: The authors thank the reviewer for this comment. In the QCMD experiment, the amount of CaCO_3 deposited on the sensor surface was very small; crystals formed are small in size ($<30 \mu\text{m}$ in diameter) and are sporadically distributed on the coated sensor surfaces (Figure 2B1-4, and Figure 3B2-3). Therefore, they can be easily removed by hydraulic shear and quick dissolution of the crystals. In the experiments using real water,

CaSO₄ (instead of CaCO₃) was used as the mineral scalant as CaSO₄ is the most common, and problematic scalant in produced water. In these experiments, a much larger mass of the scalant accumulated on the surface, forming a continuous and dense layer of scale with significant thickness (>0.2 mm on hBN and much larger on other surfaces) that covered the whole inner surface of the pipe (Figure 5). The gypsum crystals formed by CaSO₄ is also much more stable and difficult to dissolve than vaterite formed by CaCO₃. Gypsum scales are notoriously difficult to remove.

4. Figure S1 refers to a "similar transfer method." Could you please clarify what you mean here? I saw no reference to a transfer method in the text. The images used here also seem to be fairly poor quality.

Response: The authors thank the reviewer for this comment. The method used is a PMMA-assisted transfer method. We have added description of the transfer method in the supporting information.

Page 2 in the revised supporting information:

"After synthesis, graphene and hBN films were transferred to SiO₂/Si substrates using a PMMA-assisted transfer method and characterized by Raman spectroscopy, AFM, and XPS. In this method, a PMMA film was first spin-coated on the graphene or hBN grown on Cu foil. The sample was then immersed in a FeCl₃ solution to dissolve the Cu substrate. The graphene or hBN immobilized on the PMMA film was then transferred to DI water, and subsequently collected onto a SiO₂/Si substrate with the graphene or hBN side attaching to the substrate surface. Finally, acetone and IPA were used to dissolve the PMMA layer, exposing the graphene or hBN surface. The prepared graphene- or hBN-coated SiO₂/Si was dried at 60 °C and stored at room temperature before characterization."

The relatively poor quality of the SEM images in Figure S1 is caused by the low conductivity of the samples. As hBN and PVDF are not electrically conductive, it is difficult to obtain high quality SEM images at high magnification. However, besides SEM, we have used several other methods to characterize the quality of the graphene and hBN coatings. As illustrated in Fig. S2, Raman spectra, AFM images, and high-resolution XPS all show that the prepared graphene and hBN have high quality.

5. The longevity of coatings would also be of interest. Can you comment on the ability of these coatings to 1) be cleaned, 2) resist over time and 3) the general durability on surfaces?

Response: The authors thank the reviewer for this comment. The hBN- and graphene-coatings formed on quartz crystal sensors were cleaned and reused multiple times in the QCMD study. The results were reproducible, suggesting that the coatings can be cleaned after CaCO₃ scale formation. However, the effectiveness of scale removal by cleaning depends strongly on the type of mineral scalant and the cleaning method used. Gypsum, for example, is much more difficult to remove by cleaning once it forms a scaling layer.

In a supersaturated solution, scale formation is kinetically limited. A very important indicator of scale formation kinetics is the induction time. A system with a hydraulic retention time significantly shorter than the induction time can minimize scale formation as it does not allow enough time for nucleation and crystal growth in the system. The hBN

coating increases the induction time significantly by minimizing surface induced heterogeneous nucleation. In long term operation, potential damages on the coating may create additional surface nucleation sites, and therefore reduces the induction time. This could lead to decrease in scaling resistance.

Regarding durability, *h*BN is superior to graphene, PVDF, and Ti. As shown in the table below, *h*BN has a tensile strength of 68-215 GPa, the highest among the 4 materials compared, as well as very high melting point (2973 °C) and high thermal stability (~1000 °C) for use in high temperature applications. In addition, *h*BN has >10 orders of magnitude higher electrical resistivity than graphene, Ti and PVDF, which protects it from electrochemical corrosion in an electricity field. Furthermore, the interlayer integrity (binding) between *h*BN nanosheets is significantly better than graphene¹², which makes it more resistant to abrasion and hydraulic shear than graphene, and hence a more attractive candidate in challenging environment.

	Ti	PVDF	Graphene	h BN
Tensile strength	220 MPa	35-50 MPa	70–130 GPa	68–215 GPa
Melting point (°C)	1668	165-175	~3680	2973
Thermal stability (°C)	200-250	~150	~500	~1000
Resistivity (Ω · cm)	4.20×10^{-7}	1014	10^{-6}	$10^8 - 10^{13}$

Despite these superior material properties, the long-term scaling resistance and durability of *h*BN as a coating in various applications need to be evaluated under conditions that represent those in real applications. We have added text in the conclusion section of the revised manuscript to emphasize the need for future research on these issues. Please see line 354 – 358 in the revised manuscript:

*“The results of the study provide important insights for future development of novel functional materials by manipulating their interactions with surrounding media. On the other hand, the scalability, durability, and long-term scaling resistance of the *h*BN coating as well as the specific roles of substrate materials and defects in large scale coatings need to be evaluated before practical applications can be possible.”*

6. Overall, this is a nice manuscript.

Response: The authors thank the reviewer for this comment.

Reviewer #3 (Remarks to the Author):

Formation of mineral scale on a material surface has been a big problem for a variety of applications. Understanding and controlling mineral scaling behaviour is very important for R&D of next generation materials and technologies. This ms reports the superior resistance of hexagonal boron nitride (*h*BN) to mineral scale formation compared to not only common metal and polymer surfaces but also to graphene, which is the highly resistant to scaling, making *h*BN the most robust and scaling resistant material. It is a very interesting idea and an important study in the field of 2D nanomaterials.

In this ms, the detailed experimental and simulation results showed that this ultrahigh scaling-resistance is attributed to the combination of *h*BN’s atomically-smooth surface, in-plane atomic energy corrugation due to the polar boron-nitrogen bond, and

interatomic spacing that matches closely with the size of a water molecule. Furthermore, atomically hBN are normally considered to be chemically inert, which will make hBN successful and useful in the surface coating industry.

I would recommend this ms is accepted for a publication in Nature Communications after addressing the following comments.

The specific comments are:

1. In evaluation of scale formation on graphene and hBN, a supersaturated CaSO₄ solution was used, what is pH value of this solution? pH would play a role for the scale formation?

Response: The pH of the supersaturated CaSO₄ solution was 7. The reviewer is correct that pH plays an important role in the scale formation process. The specific impact of pH depends on the mineral scalant involved. In our experiment, CaSO₄ solubility is not affected by pH when the pH is higher than 2.0. We selected our solution pH to be above 2 while avoiding Ca(OH)₂ precipitation at high pH. The supersaturated CaSO₄ solution we use has an unadjusted pH of 7.0 and a saturation index of 3.2. At this pH and Ca²⁺ concentration, Ca(OH)₂ is well undersaturated, with the activity product of Ca²⁺ and OH⁻ ($\{Ca^{2+}\}\{OH^{-}\} \approx 10^{-10}$) 4 orders of magnitude lower than the Ca(OH)₂ solubility product constant ($K_{sp} = 5.02 \times 10^{-6}$). We have added the information of pH in line 385 – 387 of the revised manuscript.

“The supersaturated CaSO₄ solution (50 mmol L⁻¹) was prepared by mixing 100 mmol L⁻¹ CaCl₂ with 100 mmol L⁻¹ Na₂SO₄ at a 1:1 volume ratio. The prepared solution had a pH of 7.0 and a saturation index (SI) of 3.28.”

2. In method, saturation index was given, what does it mean? The brief introduction and refs are required.

Response: The authors thank the reviewer for this suggestion. The saturation index (SI) is defined as the ratio between the chemical activity product of the mineral ions (ion activity product, IAP) and their solubility product (K_{sp}). A SI lower than 1 means the solution is under saturated, while a supersaturated solution has a SI value greater than 1. We have added the definition of SI in line 386 – 388 of the revised manuscript. Reference has also been added in the revised manuscript.

“The prepared solution had a pH of 7.0 and a saturation index (SI) of 3.28. SI is defined as the ratio between the chemical activity product of the mineral ions and their solubility product⁶³.”

Line 696 – 698 of the revised manuscript:

“63. Sand, K. K., Tobler, D. J., Dobberschütz, S., Larsen, K. K., Makovicky, E., et al. Calcite growth kinetics: dependence on saturation Index, Ca²⁺:CO₃²⁻ activity ratio, and surface atomic structure. Cryst. Growth Des. 16, 3602-3612 (2016).”

3. The QCMD technique was used to real-time characterise scale formation, what is the sensitivity of QCMD sensor? I suggest some references should be added.

Response: The authors thank the reviewer for the comment. For the QCMD instrument (Qsense E4, Biolin Scientific, Sweden) and the 5MHz sensors used in our experiment, the typical mass sensitivity in liquid is about 1.8 ng/cm², with an optimal sensitivity of 0.5

ng/cm². The normal dissipation sensitivity is about 0.1×10^{-6} , with a maximum of 0.04×10^{-6} . This information has been added in the revised manuscript in line 413 – 416. A reference has also been added in the manuscript.

“The heterogeneous nucleation and crystal formation of CaCO₃ was also characterized in situ and in real time using the quartz crystal microbalance with dissipation (QCMD) technique (Qsense E4 analyzer, Biolin Scientific, Sweden), which has a mass sensitivity and a dissipation sensitivity of 1.8 ng cm⁻² and 0.1×10^{-6} in liquid, respectively⁶⁴.”

Line 700 in the revised manuscript:

“64. Uddenberg, H. Q-Sense E4 operator manual. Sweden: Q-Sense AB; 2009. p. 57.”

References

1. Li, C., Zhang, X., Oliveira, E. F., Puthirath, A. B., Neupane, M. R., *et al.* Systematic comparison of various oxidation treatments on diamond surface. *Carbon* **182**, 725-734 (2021).
2. Levita, G., Kajita, S., Righi, M. C. Water adsorption on diamond (111) surfaces: an *ab initio* study. *Carbon* **127**, 533-540 (2018).
3. Qu, J., Li, Q., Luo, C., Cheng, J., Hou, X. Characterization of flake boron nitride prepared from the low temperature combustion synthesized precursor and its application for dye adsorption. *Coatings* **8**, 214 (2018).
4. Li, D., Müller, M. B., Gilje, S., Kaner, R. B., Wallace, G. G. Processable aqueous dispersions of graphene nanosheets. *Nat. Nanotechnol.* **3**, 101-105 (2008).
5. Grosjean, B., Bocquet, M. L., Vuilleumier, R. Versatile electrification of two-dimensional nanomaterials in water. *Nat. Commun.* **10**, 1656 (2019).
6. Tocci, G., Joly, L., Michaelides, A. Friction of water on graphene and hexagonal boron nitride from *ab initio* methods: very different slippage despite very similar interface structures. *Nano Lett.* **14**, 6872-6877 (2014).
7. Li, H., Zeng, X. C. Wetting and interfacial properties of water nanodroplets in contact with graphene and monolayer boron-nitride sheets. *ACS Nano* **6**, 2401-2409 (2012).
8. Huang, R., Yi, P., Tang, Y. Probing the interactions of organic molecules, nanomaterials, and microbes with solid surfaces using quartz crystal microbalances: methodology, advantages, and limitations. *Environmental Science: Processes & Impacts* **19**, 793-811 (2017).
9. Huang, K., Szlufarska, I. Friction and slip at the solid/liquid interface in vibrational systems. *Langmuir* **28**, 17302-17312 (2012).
10. Johannsmann, D., Reviakine, I., Richter, R. P. Dissipation in films of adsorbed nanospheres studied by quartz crystal microbalance (QCM). *Anal. Chem.* **81**, 8167-8176 (2009).

11. Watts, E. T., Krim, J., Widom, A. Experimental observation of interfacial slippage at the boundary of molecularly thin films with gold substrates. *Physical Review B* **41**, 3466-3472 (1990).
12. Falin, A., Cai, Q., Santos, E. J., Scullion, D., Qian, D., *et al.* Mechanical properties of atomically thin boron nitride and the role of interlayer interactions. *Nat. Commun.* **8**, 1-9 (2017).
13. Sand, K. K., Tobler, D. J., Dobberschütz, S., Larsen, K. K., Makovicky, E., *et al.* Calcite growth kinetics: dependence on saturation Index, $\text{Ca}^{2+}:\text{CO}_3^{2-}$ activity ratio, and surface atomic structure. *Cryst. Growth Des.* **16**, 3602-3612 (2016).

REVIEWERS' COMMENTS:

Reviewer #1 (Remarks to the Author):

The authors have successfully addressed all my comments. I recommend the paper for publication.

Reviewer #2 (Remarks to the Author):

I think the authors have an excellent manuscript. I have no additional comments on this work.

Reviewer #3 (Remarks to the Author):

My comments and questions were well addressed.

I would recommend this revised ms is accepted for a publication in Nature Communications.

REVIEWERS' COMMENTS

Reviewer #1 (Remarks to the Author):

1. The authors have successfully addressed all my comments. I recommend the paper for publication.

Response: The authors thank the reviewer for the recommendation. Your helpful comments have greatly improved the quality of our manuscript.

Reviewer #2 (Remarks to the Author):

1. I think the authors have an excellent manuscript. I have no additional comments on this work.

Response: The authors thank the reviewer for the recommendation. Your helpful comments have greatly improved the quality of our manuscript.

Reviewer #3 (Remarks to the Author):

1. My comments and questions were well addressed. I would recommend this revised ms is accepted for a publication in Nature Communications.

Response: The authors thank the reviewer for the recommendation. Your helpful comments have greatly improved the quality of our manuscript.